# Antiviral Activity of Rhamnolipids Nano-Micelles Against Rhinoviruses—In Silico Docking, Molecular Dynamic Analysis and In-Vitro Studies

**DOI:** 10.3390/cimb47050333

**Published:** 2025-05-06

**Authors:** Lila Touabi, Nasser S. M. Ismail, Marwa R. Bakkar, Gary R. McLean, Yasmin Abo-zeid

**Affiliations:** 1School of Human Sciences, London Metropolitan University, 166-220 Holloway Road, London N7 8DB, UK; l.touabi@londonmet.ac.uk; 2Pharmaceutical Chemistry Department, Faculty of Pharmacy, Ain Shams University, Abassia, Cairo 11566, Egypt; nasser.mohamed@pharma.asu.edu.eg; 3Botany and Microbiology Department, Faculty of Science, Helwan University, Ain Helwan, Cairo 11795, Egypt; marwa_mahmoud01@science.helwan.edu.eg; 4National Heart and Lung Institute, Imperial College London, Norfolk Place, London W2 1PG, UK; 5Department of Pharmaceutics and Industrial Pharmacy, Faculty of Pharmacy, Helwan University, Ain Helwan, Cairo 11795, Egypt

**Keywords:** rhinoviruses, rhamnolipids nano-micelles, antiviral agent, docking study

## Abstract

Hospital-acquired infections (HAIs) previously focused mainly on multidrug-resistant (MDR) bacteria, with less attention on viruses. The COVID-19 pandemic highlighted the importance of controlling viral infections. Human rhinoviruses (HRVs) are among the viruses responsible for HAIs. HRVs are non-enveloped viruses that infect the upper airways after airborne or direct transmission. Due to their lack of a membrane envelope, HRVs exhibit moderate resistance to commonly applied alcoholic disinfectants. Therefore, there is a significant need to develop alternative disinfection and hand sanitation strategies to control HRV infections in healthcare settings without posing a risk to human health. The antimicrobial activity and safety of rhamnolipids and rhamnolipids nano-micelles (RMN) against MDR-bacteria and several viruses, including SARS-CoV-2, were confirmed recently. Also, we previously demonstrated the superior antimicrobial activity of RMN over rhamnolipids. In the current study, molecular docking demonstrated the weak interactions of rhamnolipids with HRV-1A (minor group) compared to HRV-14 (major group), suggesting a superior antiviral activity of rhamnolipids towards major group rhinoviruses. To biologically validate these data, RMN was prepared and characterized, and then antiviral activity against HRV-16 (major group) and HRV-1B (minor group) infection of HeLa cells was assessed. RMN showed a complete inhibition of HRV-16 infection with recovery of 100% of HeLa cell viability. In contrast, only partial inhibition of HRV-1B infection with approximately 50% protection against infection was observed. Therefore, RMN might be recommended as a disinfectant and/or a hand sanitizer component to control the spread of RVs in hospital care settings or elsewhere to reduce the incidence of respiratory infections.

## 1. Introduction

Hospital-acquired infections (HAIs) have long been a major concern, primarily due to multidrug-resistant bacteria (MDR) [1,2,3,4], with less focus on viral infections. However, the COVID-19 pandemic has shifted attention toward controlling viral infections in hospital settings [5], especially in Neonatal Intensive Care Units (NICU) and Pediatric Intensive Care Units (PICU). Rhinoviruses (RVs) are among the viruses known to cause nosocomial respiratory infections in these settings [6].

RVs primarily target the airway epithelium and are considered the main cause of the common cold [7]. RV infections of the respiratory system are also linked to acute exacerbations of asthma and chronic obstructive pulmonary disease (COPD) [8,9,10,11]. Additionally, RVs are the second most common virus, after respiratory syncytial virus, responsible for viral bronchiolitis. They have also been implicated in causing sinusitis and lower respiratory tract diseases [12,13,14].

RVs are non-enveloped viruses with a positive-sense single-stranded RNA genome. This genome is encased in an icosahedral protein capsid, which is composed of 60 copies of each of the four viral capsid proteins: VP1, VP2, VP3, and VP4 [15]. The surface of the RV capsid, made up of repeating units of the VP1-VP2-VP3 protomer, is extremely variable among the approximately 160 serotypes. Based on genome sequences, RVs are divided into three groups, referred to as A, B, and C, with types A and B further classified as major or minor groups based on entry receptor usage. Specific capsid regions of RV serotypes form contacts with one of the three defined entry receptors expressed by airway epithelial cells to facilitate access to the host cell by receptor-mediated endocytosis [16]. Once inside cells, the virion uncoats to reveal the RNA genome, beginning the replication process to make new virus particles. The development of antivirals for RVs has proved challenging due to the variable nature of the capsid and the paucity of druggable viral enzymes (just protease and polymerase). Broad-spectrum antivirals have shown limited abilities to inhibit multiple RV serotypes but drugs that bind the capsid hydrophobic pocket, shared amongst most RV types, have achieved better success [17]. However, both pleconaril and vapendavir have failed clinical trials due to off-target effects and lack of efficacy. New approaches are therefore required.

The transmission of RVs typically occurs through direct exposure by inhalation of respiratory droplets or micro-droplets. It can also spread via fomites, such as contaminated surfaces and inanimate objects, or through direct person-to-person contact [18,19]. Due to the absence of a membrane envelope, RVs exhibit moderate resistance to common disinfectants like alcohol hand rubs [20]. Consequently, there is a pressing need to identify alternative disinfectants and hand sanitizers that can effectively inactivate RVs without causing harm to healthcare providers or hospitalized patients in healthcare settings.

Rhamnolipids are biosurfactants economically produced by *Pseudomonas aeruginosa*, a bacterial source with potential for scale-up to industrial production levels [21,22,23]. The broad-spectrum antibacterial activity of rhamnolipids [24,25,26,27] and its compatibility with animal skin have been previously reported [28]. Additionally, rhamnolipids have demonstrated antiviral activity against various viruses, including a clinical isolate of Severe Acute Respiratory Syndrome Coronavirus-2 (SARS-CoV-2) strain VR PV10734, human coronaviruses 229E (ATCC VR-740) and OC43 (ATCC VR-1558), herpesviruses HSV-1 strain SC16, HSV-1 (GFP-HSV-1) and HSV-2 strain 333, and the non-enveloped picornavirus Poliovirus Type 1 (PV-1) strain Chat (ATCC VR1562) [29,30]. Our recent studies have shown that enhancement of the antibacterial and antiviral activity of rhamnolipids can be achieved by their formulation into nano-micelles (RMN), while also ensuring compatibility with animal skin and eyes [1,5].

To the best of our knowledge, the antiviral activity of RMN against RVs has not been explored. In this study, we investigate the in vitro antiviral activity of RMN against two distinct RV serotypes, HRV-16 and HRV-1B. These serotypes of RVs were selected for this study because they are classified as level II pathogens and represent the major (A16) and minor (A01) groups of RVs, respectively. These two serotypes enter host cells via different cell-surface receptors (ICAM-1 for the major group; VLDLR for the minor group) and undergo distinct uncoating processes within the cells through endocytic mechanisms [31]. While both major and minor group RVs are believed to enter the endocytic pathway [32], they are thought to exit endosomes through different mechanisms [31]. HeLa cells, a human cervical carcinoma cell line, are commonly used for the propagation of RVs [33]. They are ideal for the current study as they can be infected by both major and minor group RVs [34] owing to their expression of the major group RV receptor, ICAM-1, and the minor group RV entry receptor, LDLR [35].

In this study, docking and molecular dynamic simulation analyses were first conducted to evaluate potential interactions (if any) between rhamnolipids and the capsid of major and minor group rhinoviruses, specifically HRV-14 and HRV-1, respectively. Secondly, RMN was prepared from bacterial-produced rhamnolipids, characterized, and first assessed for cytotoxicity against HeLa cells. Third, the ability of RMN to inhibit the infectivity of HRV-16 and HRV-1B in vitro was then investigated. Based on the findings, RMN is recommended for use as antivirals for RVs and may find utility as disinfectants and/or hand sanitizers in healthcare and other high-risk settings to control the spread of RVs, reducing the incidence of respiratory infections.

## 2. Materials and Methods

### 2.1. Materials

Rhinovirus stocks and HeLa cells were kindly provided by Prof S Johnston, National Heart and Lung Institute, Imperial College London. DMEM medium, fetal calf serum (FCS), non-essential amino acids (NEAA), penicillin, and streptomycin were purchased from Thermo Fisher Scientific, Paisley, UK.

Microbiological media (Tryptic soy broth; TSB, tryptic soy agar; TSA and Muller Hinton agar; MHA) were purchased from HiMedia (HiMedia, Mumbai, India). Peptone and sodium chloride were purchased from (Oxoid, Cheshire, UK). Hydrochloric acid, ethyl acetate, and sulphuric acid were purchased from Honeywell (Honeywell™, Charlotte, NC, USA). L-rhamnose was purchased from (Sigma-Aldrich, Cairo, Egypt), and Orcinol was obtained from (SDFCL, Kolkata, India). All other materials and reagents were of analytical grade purchased from Sigma-Aldrich and used as supplied.

### 2.2. Methodology

#### 2.2.1. Computational Studies

##### In Silico Docking

We used the crystal structure of the RV capsid in complex with a known effective antiviral agent as a reference to validate the docking study of rhamnolipids interacting with viral proteins for both minor and major groups of RVs. Accordingly, the crystal structure of the HRV-14 canyon floor (major group) complexed with an antiviral compound, known as R 61837, was obtained from the Protein Data Bank (PDB: https://doi.org/10.2210/pdb1R09/pdb, accessed on 3 December 2024) [36] to assist with the docking studies involving interactions of HRV-14 canyon floor with rhamnolipids. Additionally, the crystal structure of the HRV-1A canyon floor (minor group) was obtained (PDB: https://doi.org/10.2210/pdb1R1A/pdb, accessed on 3 December 2025) and its structure in complex with the antiviral compound WIN53338, reported in the literature [37], was used to assist with docking studies involving interactions between HRV-1A and rhamnolipids.

To ensure the reliability of our docking results, proteins were prepared by cleaning, adding hydrogen atoms, and applying CHARMm charge and MMFF partial charge force fields. The structures were subsequently minimized, and binding sites were defined as reported [38]. Docking results were selected based on both energy scoring and referenced clusters to ensure statistical representativeness.

The rhamnolipids molecules were drawn using ChemDraw V.14 and saved as mol extension for further view on Discovery Studio Software. The compounds were simulated with similar forcefields to the protein and then prepared as ligands. Docking procedures were run adopting the C-Docker protocol within Discovery Studio 4.0 using the same forcefields applied previously. The results of interaction energies were sorted and recorded together with the binding modes of compounds and were viewed via 3D view modes.

##### Standard Dynamic Simulations

The dynamic simulation studies were performed using Discovery Studio V. 4.0 and applied to free protein and rhamnolipids molecules. Standard Dynamic Cascades was applied where the first minimization algorithm was set to steepest descent with maximum steps 2000 and RMS gradient 1.0. The second minimization algorithm was set to conjugate gradient with maximum steps 1000. The initial temperature was set to 50, and the target temperature was 300 °C with a maximum velocity of 2000. On the other hand, the equilibration phase was set with a simulation time of 10 Ps and an interval of 2 Ps. The Implicit Solvent Model was set to Generalized Born with Simple Switching (GBSW), and the dynamics integrator protocol used Leapfrog Verlet [39]. During molecular dynamics simulations, a single long-timescale simulation was performed rather than multiple independent runs. Therefore, the results presented are based on this continuous trajectory, rather than an average of multiple simulations. The receptor structure was obtained from a reliable crystallographic source, which minimizes the need for additional equilibration. In addition, two minimization steps 1 and, 2 were carried out for protein during the molecular dynamic simulation (standard Dynamic cascade protocol) before the production step.

#### 2.2.2. Production and Characterization of Rhamnolipids

Rhamnolipid production was performed using the shake-flask technique following our previously reported protocol [1,5,40] with purification conducted as previously reported [41]. The crude rhamnolipid extract was characterized using ESI-MS analysis in both positive and negative ion modes on an XEVO TQD triple quadrupole mass spectrometer (Waters Corporation, Milford, MA 01757, USA). The obtained peaks and spectra were processed using MassLynx 4.1 software, following our previously published protocol [40].

#### 2.2.3. Preparation and Characterization of RMN

RMNs were prepared following our previously published protocol [40]. In brief, rhamnolipids (1 mg/mL) were sonicated in phosphate-buffered saline (PBS, 10 mM, pH 7.4) or DMEM using a probe sonicator (Dr. Hielscher Sonicator, Teltow, Germany) to form the nano-micelles. Malvern Zeta-sizer Nano ZS (Malvern Instruments, Ltd., Malvern, Worcestershire, UK) was used to determine the particle size, polydispersity index (PDI), and zeta potential of the formed nano-micelles, at 25 °C ± 0.1. For relevant results, RMN samples were diluted in PBS (10 mM, pH 7.4) to give a count rate range of 50 to 300 KCPs.

#### 2.2.4. Cytotoxicity of RMN

##### Cell Culture

HeLa cells were seeded in a T175 flask containing DMEM supplemented with 10% FCS, and 1% penicillin-streptomycin solution, and incubated at 37 °C with 5% CO_2_. The culture medium was refreshed every 24 h until the cells reached 70–90% confluency, at which point they were sub-cultured. For sub-culturing, the medium was removed, and the cells were washed twice with PBS. Trypsin/EDTA was then added to detach the cells from the flasks, followed by suspending the cells in fresh complete medium (DMEM with 10% FBS, 1% penicillin-streptomycin, 1% l-glutamine, and 1% non-essential amino acids). The cells were collected by centrifugation at 1500 rpm and adjusted to the required density for subsequent experiments.

##### RMN Cytotoxicity on HeLa Cells

The assay was conducted following our previously published protocol [42] with modifications. Briefly, HeLa cells (2 × 10^5^ cells/well) were seeded into 96-well culture plates and supplemented with 100 µL of DMEM containing 10% FCS and 1% penicillin-streptomycin solution. The cells were incubated at 37 °C with 5% CO_2_ for 24 h. After removing the culture medium, 100 µL of RMN, serially diluted in PBS (10 mM, pH 7.4) or DMEM (2% FBS), was added to the wells and incubated for 72 h under the same conditions. After incubation, the RMN-containing media were removed from the adherent cells’ monolayer and the adherent cell monolayer was gently washed three times by filling and emptying each well of the plate with 200 µL of PBS using a multichannel pipette. Next, 50 µL of crystal violet (CV, 0.1% *w*/*v*) solution was added to each well and incubated at room temperature for 5 min. The stained monolayers were again gently washed 4–5 times but this time with double-distilled water, followed by drying by inverting the plate for 1 h. Finally, the crystal violet stain was solubilized using 1% sodium dodecyl sulfate (SDS). The cytotoxicity of RMN was expressed as the percentage of viable cells, calculated by dividing the optical density of the treated sample by that of the control sample, measured at 560 nm using a FLUOstar Omega plate reader (software: V6.20 edition 4 with MARS data analysis V5.02 R3) and multiplying by 100. Control samples included untreated HeLa cells, wells with media and no cells, wells containing PBS (10 mM, pH 7.4, the vehicle used for RMN dispersion), all incubated under the same conditions [43]. Two independent experiments were conducted, with each sample prepared in triplicate.

#### 2.2.5. RVs Propagation and Tissue Culture Infectious Dose 50% (TCID50) Determination

TCID_50_ of RV preparations was determined following our previously published protocol [42]. HeLa cells are routinely used for RV propagation [44] and this approach consistently yields virus titers (TCID_50_/mL) of 3–4 × 10^7^ for both HRV-16 and HRV-1B according to our previously published methods [42]. Briefly, HeLa cells were cultured in T175 flasks with DMEM supplemented with 10% FCS and 1% penicillin-streptomycin until reaching 80% confluence. The culture medium was removed, and the cells were infected with RVs preparations diluted in DMEM with 2% FCS. The flask was gently shaken for 1 h at room temperature to enhance virus attachment, followed by incubation at 37 °C with 5% CO_2_ for 72 h until a cytopathic effect (CPE) greater than 80% was observed. Cells were then subjected to three freeze-thaw cycles and briefly vortexed to lyse cells and release the cell-associated virus. The viral solution was clarified by centrifugation at 4000 rpm for 15 min, after which the supernatant containing the virus was collected, aliquoted, and stored at −80 °C. The RV stock solution was titrated to determine infectious RV levels by infecting a HeLa cell monolayer (5 × 10^4^ cells/well) in a 96-well microtiter plate with serially diluted virus (6 wells each at 10^−1^ to 10^−9^) in DMEM supplemented with 2% FCS and 1% penicillin-streptomycin. Plates were incubated at 37 °C with 5% CO_2_ for three days, after which wells were examined microscopically for the presence or absence of a cytopathic effect (CPE) to determine the 50% tissue culture infectious dose (TCID_50_). This value represents the final virus dilution that shows CPE in 50% of the cultured wells.

#### 2.2.6. Antiviral Activity of RMN

HeLa cells were seeded into 96-well microtiter plates at a density of 2 × 10^4^ cells/well in DMEM supplemented with 10% FBS, 1% penicillin-streptomycin, 1% l-glutamine, and 1% non-essential amino acids, then incubated at 37 °C with 5% CO_2_ for 24 h. RVs (50 µL, 100 TCID_50_/mL, MOI < 0.01) were mixed with 50 µL of RMN solution, which had been serially diluted in PBS (10 mM, pH 7.4) or DMEM (2% FBS), and incubated at room temperature for 1 h under gentle shaking. The RMN/RVs mixture (100 µL) was then added to the seeded HeLa cells where growth media had been removed, and the final volume in each well was adjusted to 200 µL with DMEM (2% FBS) or PBS (10 mM, pH 7.4). The plate was incubated for three days, after which the media were removed, and cell viability was assessed using crystal violet (CV) staining as described in Section 2.2.4. A set of control samples was prepared, including a negative control for RMN, cells treated with the vehicle used to disperse RMN, (PBS, 10 mM, pH 7.4) or DMEM (2% FBS). The infection-only control consisted of HeLa cells cultured in DMEM (2% FBS) and infected with HRV-16 or HRV-1B. Two independent experiments were performed, with each sample prepared in triplicate.

#### 2.2.7. Statistical Analysis

All statistical analysis was performed using two-way ANOVA followed by a post-hoc Tukey test. Analyses were carried out using GraphPad Prism 9.0 software at a confidence level (95%).

## 3. Results and Discussion

### 3.1. In-Silico Studies

For this study, we required RV capsid structures complexed with antivirals bound in the hydrophobic pocket (canyon), these were available for RV-1A and RV-14 as they were the prototype RVs extensively studied in the late 1980s and early 1990s. For biological studies, our laboratory routinely uses RV-1B and RV-16 for in vitro experiments, as they are both capable of infecting HeLa cells, are used for infection of mice [45], and RV-16 is used widely for experimental human challenge (Mallia). The capsid sequences of RV-1A and RV-1B are very closely related (Table 1), so the use of RV-1A structural data can be easily extrapolated to the biological experimentation using RV-1B. The comparison between RV-14 and RV-16 is more complex, however, as these serotypes belong to different groups A and B (Table 1). Nevertheless, enough similarity exists within the hydrophobic pocket to compare the RV-14 structural data to that obtained experimentally using RV-16.

The antiviral compounds such as R 61837 and WIN 53338 were reported to interact with amino acids within the hydrophobic pocket in areas of VP1 that lie on the canyon floor for both minor (HRV-1A) and major (HRV-14) groups of rhinoviruses. Such interactions lead to capsid conformational changes that can interfere with infectivity [37]. In the major group, these conformational changes prevent the virus from binding to host cell receptors, resulting in virus inactivation [37]. However, with the minor group RVs, less pronounced conformational changes were reported, allowing the virus to attach to the receptor but still blocking infectivity suggesting an alternate mechanism [37].

Based on previous studies [36,37,38], to check the potential antiviral activity of rhamnolipids against both major (HRV-14) and minor groups (HRV-1A), both mono and di-rhamnolipids were docked to VP1 residues near to or part of the canyon floor as previously defined [36,37,38]; to identify the incidence of any conformational changes similar to what previously reported to occur with the antiviral compounds; R 61837 and WIN 53338.

Individual mono- and di-rhamnolipid molecules, rather than nano-micelles, were used in the docking study, as previous findings indicated that rhamnolipids, both in their molecular form and as nano-micelles can inactivate SARS-CoV-2 [5,40].

#### 3.1.1. Molecular Docking Study

A molecular docking study was applied using the C-Docker protocol in Discovery Studio 4.1 Software. Re-docking the lead compound (R 61837) in the HRV-14 active site. This was followed by the alignment of the X-ray bioactive conformer of the lead compound with the best-fitted pose achieved from the docking run. The alignment showed good coincidence between them with RMSD = 0.21A°, indicating the ability of the used docking protocol to retrieve valid docking poses (Appendix A) [46,47]. The mono and di-rhamnolipids were prepared and docked into VP1 residues lying on the canyon floor of HRV-14 and HRV-1A [36,37,38]. The analysis of the binding modes of rhamnolipids was applied to obtain more details about the binding and interactions of rhamnolipids to the key amino acids of the canyon floor of HRV-14 and HRV-1A and therefore, to help interpret rhamnolipids anti-RV mechanisms of action [36,37,38].

Concerning HRV-14 (major group); the docking study of rhamnolipids with VP1 residues lying on the canyon floor showed conformational changes at the HRV-14 canyon floor (Table 2). This would likely result in RV capsid inactivation via the inability of the virus to bind with host cell ICAM-1 receptor. The X-ray crystallographic structure of the HRV-14 canyon floor complexed with an antiviral compound, R 61837 reported an interaction with four amino acids in that location [36]. These four canyon floor amino acids include Lys103, Pro155, His220, and Ser223 and are reported to be essential to bind with host cell ICAM-1 receptor [38]. The binding modes of rhamnolipids with the above-named essential amino acids and the values of interaction energy are presented in Table 2. The molecular docking of the rhamnolipids using C-Docker protocol revealed that mono-rhamnolipids retained four hydrogen-bonds (4H-bonds) with Lys103 (1HBA), Leu106 (2HBA) and Ser107 (1HBD) and showed hydrophobic interactions with Leu218, Cys199 and Ser223 where the interaction energy (−71.25 Kcal/mol) was lower than the antiviral compound, R 61837 (−55.5 Kcal/mol). This indicates the formation of a more stable complex with mono-rhamnolipids than the complex formed with R 61837. Di-rhamnolipids were stacked at the binding site via a single hydrogen bond (1HBA) with His220 and showed hydrophobic interactions with the essential amino acids; Lys103, Pro155, and Ser223 where the interaction energy was −70.11 Kcal/mol, that is lower than the interaction energy identified with R 61837 (−55.5 Kcal/mol) (Table 2). The interaction energy reported for R 61837 interaction with the VP1 residues located at the canyon floor is attributed to a single hydrogen bond (1HBA) with Lys 103 and hydrophobic interactions with Leu106, Tyr152, Ser223, and His220. Taken together, the obtained data indicated a more effective interaction of rhamnolipids with the VP1 residues lying at the canyon floor of HVR-14 compared to R 61837, suggesting that rhamnolipids may interfere with major group RVs binding host cell ICAM-1 receptor, resulting in loss of virus infectivity.

For minor group RV, a docking study of mono and di-rhamnolipids with VP1 residues lying on the canyon floor of HRV-1A was carried out using the X-ray crystallographic structure of HRV-1A (PDB ID: PDB: https://doi.org/10.2210/pdb1R1A/pdb, accessed on 3 December 2024) and its complex structure previously reported with the antiviral compound, WIN 53338 that effectively inactivates HRV-1A [37]. Docking studies performed for the validation purpose tracking the interactions of WIN 53338 with the amino acids located at the canyon of HRV-1A [37] showed a single hydrogen bond (1HBA) formation with Asn215 and hydrophobic interactions with Ile101, Leu103, Ile125, Tyr147, PIle171, Pro185, Leu187, Ile220, Ile 184, Tyr198, Met217, His268 with interaction energy equivalent to 61.25 Kcal/ mol (Table 3). Next, the binding mode of rhamnolipids with these essential amino acids and the values of interaction energy were assessed and presented in Table 3. The docking study revealed that mono-rhamnolipids retained hydrophobic interactions only with Tyr145, Tyr147, Ile171, Phe182, Pro185, and Ile220 where the interaction energy was −50.40 Kcal/mol. Meanwhile, di-rhamnolipids form one hydrogen bond (1HBA) with Met169 and hydrophobic interactions with; Tyr147, Pro149 Ile171, Phe182, Pro185 and Ile220 where the interaction energy was −53.71 Kcal/mol (Table 3). Thus, both mono and di-rhamnolipids form a more stable complex with the essential amino acids at the canyon floor compared to WIN 53338. In other words, rhamnolipids were able to interact more effectively with these essential amino acids than the antiviral compound, WIN 53338, and therefore, these interactions might be associated with a conformational change that could stabilize capsid structure, preventing uncoating and resulting in loss of virus infectivity.

However, to assess the antiviral efficacy of mono and di-rhamnolipids based on their interactions with the essential amino acids involved in the binding of HRV-14 and the stabilization of HRV-1A capsid, the interaction energy between rhamnolipids and viruses were compared. As presented in Table 2 and Table 3, the interaction energy recorded for HRV-1A interaction with mono and di-rhamnolipids was higher than the interaction energy values recorded for HRV-14. This implied that the rhamnolipids-HRV-1A complexes are more unstable than the rhamnolipids-HRV-14 complex, indicating a less effective interaction of rhamnolipids with the HRV-1A (minor group) compared to that of HRV-14 (major group). Therefore, a less efficient HRV-1A inactivation might occur compared to HRV-14 with rhamnolipids. Additionally, a molecular docking study was performed after dynamic simulation for the best poses of mono and di-rhamnolipids into HRV-14 and HRV-1A. Docking results for both mono and di-rhamnolipids after molecular dynamic (MD) simulation revealed additional molecular interactions with the key amino acids in the active site of both HRV-14 and HRV-1A as shown in Table 2 and Table 3.

#### 3.1.2. Standard Dynamic Simulation

Since molecular docking represented only a single snapshot of protein-ligand interactions, molecular dynamic simulations were performed in order to study the protein-ligand interactions in motion contributing to their stable bound conformation and to visualize the effect of ligand binding (e.g., rhamnolipids) on the conformation of the protein (e.g., VP1 residues located at virus canyon). Initially, the free protein structure of VP1 residues located on the canyon floor of HRV-14 and HRV-1A was subjected to molecular dynamic (MD) simulations, then after MD simulations for HRV-14 canyon and HRV-1A complexed with rhamnolipids were performed and assessed.

Concerning major group HRV-14, the total energy versus time (Figure 1A), root-mean-square deviation (RMSD) (Figure 2A), and root mean square fluctuations (RMSF) for free protein structure of VP1 residues located at the canyon of HRV-14 (Figure 3A) were studied. Contributions were plotted as a time-dependent function of MD simulations. RMSD of the protein backbone was found to fluctuate around 11.25 Å (Figure 2A). These averaged constant graphs suggested that the protein structure was stable throughout each simulated time of 200 ns. RMSF of all the residues was calculated during the 200 ns MD simulations to identify the higher flexibility regions in the protein. In the RMSF graph of HRV-14 (Figure 3A), the major peaks of fluctuations have been observed within the initial 20 residues with over 22.5 Å, residues between 60 and 160 with up to 8.5 Å, residues between 80 and 220 with up to 6 Å and residue 240 with 9 Å. Residue 270 with 19.5 Å.

MD simulations were also performed for HRV-14 complexed with rhamnolipids to understand the effect of rhamnolipids binding with HRV-14. The best conformation pose of HRV-14 complexed with mono rhamnolipids (−71.25 Kcal/mol), and di-rhamnolipids (70.11 Kcal/mol) presented in Table 2, were used for carrying out MD simulations. The obtained results of the MD simulations are presented in Figure 1, Figure 2 and Figure 3. The results in Figure 2B,C showed that the RMSD of the trajectories of rhamnolipids complexed with HRV-14 were similar with no observed deviation from that of free HRV-14 canyon protein structure. These results are highly supportive of forming a stable HRV-14-rhamnolipids complex due to the interaction of rhamnolipids with the amino acids in VP1 residues located within the canyon structure of HRV-14, previously reported to be involved in capsid and host cell attachment. The interactions of rhamnolipids are believed to be associated with conformational changes of local polypeptides and therefore, reducing the ability of HRV to bind to host cell receptors as previously reported with the antiviral agent, R 61837 [36,37,38], resulting in capsid inactivation and loss of virus infectivity. The simulation time of 200 ns that was applied in the present study was sufficient for the side chains re-arrangement in free virus protein structure (free protein structure of VP1 residues located at the canyon structure of HRV-14) as well as virus protein–rhamnolipids complex to facilitate the most stable binding conformation. Furthermore, RMSF was also calculated for each residue index to evaluate the flexibility of the HRV-14-rhamnolipids complex. A considerable reduction of fluctuations (RMSF) compared to the free virus protein structure was recorded, indicating an enhanced stability of the complex structure of rhamnolipids and HRV-14 compared to the free virus protein structure (Figure 3B,C). Moreover, by analyzing the total energy of the stabilized conformation of virus protein structure complexed with mono and di-rhamnolipids, the average total energy of HRV-14-mono and di-rhamnolipids complex ranged from −8.300 to −8.500 and from −8.375 to −8.450 kcal/ mol, respectively (Figure 1B,C), which is lower than the value recorded for free protein structure of HRV-14 (−8.375 to −8.600 kcal/mol). Again, this indicates the stability of the complex structures.

For HRV-1A, we analyzed the energy of the stabilized conformation for HRV-1A, and the obtained data revealed that the average energy (Figure 4A) recorded was ranging from −7.950 to −8.300 kcal/mol. RMSD value of free protein structure of VP1 residue located at the canyon of HRV-1A, (Figure 4B) was found to fluctuate around 15 Å. These averaged constant graphs indicated that the protein structure was stable throughout each simulated time of 200 ns. In the RMSF graph of HRV-1A (Figure 4C), the major peaks of fluctuations were observed within the initial 40 residues with over 21 Å, residues between 80 and 220 up to 10.0 Å, residue 260 with 22.5 Å. MD simulations of HRV-1A complexed with di-rhamnolipid were carried out to assess the effect of rhamnolipids binding to amino acids of analogous VP1 residues located at the canyon structure of HRV-1A that were reported to be involved in the binding of major group HRV to host cell receptors [37]. HRV-1A complexed with di-rhamnolipid was employed for MD simulations as its complex with HRV-1A is more stable than that of mono-rhamnolipids due to having a lower total interaction energy compared to mono-rhamnolipids (Table 3). MD simulations presented in Figure 4, revealed that RMSD (Figure 4B) for HRV-1A complexed with di-rhamnolipids has a higher fluctuation value than the value recorded for the free protein structure (VP1 residues located at the canyon structure of HRV-1A), with a fluctuation value around 25 Å. These results are highly supportive of the weak interaction capability of rhamnolipids with HRV-1A and suggest either inefficient or partial inactivation of the capsid. In addition, RMSF (Figure 4C) was also calculated for each residue index to evaluate the flexibility of HRV-1A-di-rhamnolipids complexes. These showed considerable increases in fluctuations compared to that of the free protein structure (VP1 residues located at the canyon structure of HRRV-1A). This indicates the lower stability of HRV-1A-di-rhamnolipids complexes compared to the free protein structure. The average total energy of the stabilized conformation of di-rhamnolipids complexed with HRV-1A ranged from −8.050 to −8.200 kcal/mol of energy which is very similar to the average total energy recorded for the free protein structure; −7.9 to −8.3 kcal/mol (Figure 4A). Taken together, results obtained from MD simulations further demonstrated the weak interactions of rhamnolipids with HRV-1A compared to HRV-14 and this suggests a superior antiviral activity of rhamnolipids towards major group RV than minor group RV.

To ascertain the results obtained from docking studies, the antiviral activity of rhamnolipids in its nanoforms was investigated against HRV-16 (major group) and HRV-1B (minor group) in vitro, RMN was prepared, characterized and used instead of rhamnolipids due to their reported biosafety, both in-vitro and in-vivo [1,5] as well as previous identified superior antiviral activity against SARS-CoV-2 compared to rhamnolipids [5,29,30].

### 3.2. Production of Rhamnolipids

Rhamnolipids production was successfully confirmed using an ESI-MS spectrometer coupled with UPLC (LC/ESI-MS) as shown in our previous publication [40] where rhamnolipids obtained were composed of a higher proportion of mono-rhamnolipids to di-rhamnolipids.

### 3.3. Characterization of Prepared RMN

RMN colloidal dispersion was previously prepared in our lab [1] at concentrations ranging from 0.0195 to 10 mg/mL. We previously reported [1] that sample dilution was accompanied by a non-significant (*p* > 0.05) increase in particle size. Particle size ranged from 166 ± 21.7 to 265 ± 33.56 nm, with a polydispersity index (PDI) between 0.25 to 0.46, and zeta potential above −30 mV, indicating stability of nano-micelles.

### 3.4. Cytotoxicity of RMN on HeLa Cells

RMN were first assessed for their cytotoxicity on HeLa cells as explained in Section 2.2.4 and to determine the maximum safe concentrations available for antiviral assays using HeLa cells. Cytotoxicity of RMN dispersed in PBS (Figure 5A) and DMEM (Figure 5B) on HeLa cells, as measured by crystal violet staining of monolayers, was concentration dependent where an increase of RMN concentration above 60 μg/mL was accompanied by a decrease of cell viability. However, RMN dispersed in PBS (10 mM, pH 7.4) showed a greater capacity to reduce cell viability than that when dispersed in DMEM (2% FBS) between 60 and 1000 µg/mL. At RMN concentrations ≥ 200 µg/mL, the cell viability percentage recorded for DMEM (2% FBS) ranged from 79% to 15% whereas no cell viability was recorded for PBS (10 mM, pH 7.4).

Concerning RMN dispersed in PBS, a concentration of 125 µg/mL was accompanied by a medium cytotoxic effect where the cell viability percentage was approximately 36%. Further dilution of RMN concentrations (62.5–0.48 µg/mL), resulted in minimal effects on cell viability with percentages observed similarly to what was recorded for untreated HeLa cells. The data obtained in the current study was similar to the cytotoxicity of RMN on human dermal fibroblast cells we previously reported [1] where an increase of RMN concentration was accompanied by a decrease in cell viability percentage, however, the cell viability percentage recorded ranged from 60 to 100% for RMN concentrations; 312 to 19.5 µg/mL [1].

The lower viability percentage recorded with PBS compared to the culture medium is consistent with the literature [48]. PBS lack nutrients; besides it does not have significant buffer capacity, and upon dilution with PBS, an increase of shearing stress on cells might occur. Furthermore, an increase in the incubation time of cells with PBS is accompanied by desorption of shearing protectors that are present at the culture medium from the cell membrane and therefore, the cells become more sensitive to shear stress as the incubation time increases. In addition, cell survival factors and additional nutrients provided by the FBS are absent or reduced when adding PBS to the cells. This might be responsible for an increase in cell cytotoxicity of RMN when PBS was used instead of culture medium, and the observed toxicity was concentration dependent where a higher toxicity was recorded at RMN concentration > 62.5 µg/mL. Nevertheless, we established with this assay that concentrations of RMN below 60 μg/mL are acceptable for further experimentation in antiviral assays using HeLa cells as they do not contribute to decreased cell viability induced by HRV infection.

### 3.5. Antiviral Activity of RMN, Virus Infection Neutralization Test

The antiviral activity of RMN against HRV-16 and HRV-1B was performed as explained in Section 2.2.5 and results were presented in Figure 6 and Figure 7. For the antiviral activity of RMN dispersed in PBS (10 mM, pH 7.4), the results showed that protection from virus infection was only recorded with RMN concentration 125 µg/mL for both HRV-1B and HRV-16 (Figure 6A,B). Thus, for HRV-1B (Figure 6A), the best protection against virus infection was recorded with 125 µg/mL RMN where the cell viability percentage was 64%, higher than the cell viability percentage (36%) recorded for cells incubated with the same concentration of RMN (125 µg/mL) without the addition of RV (Figure 5A). The reduction of cell viability percentage compared to untreated HeLa cells (around 100%) might be attributed to the ineffective interaction of RMN with HRV-1B which might be associated with a partial inactivation of the virus capsid. Thus, a partially inactivated virus can likely attack cells and impart its cytopathic effect. By using higher concentrations of RMN (>125 µg/mL), a decrease in cell viability percentage was observed and this can be attributed to the cytotoxic effect of RMN on HeLa cells (Figure 5A). Upon using concentrations of RMN < 125 µg/mL, the reduction of cell viability percentage is likely due to the cytopathic effect of the virus as the number of RMN molecules is insufficient to inactivate viruses in the cell culture medium. Furthermore, RMN was dispersed in PBS which is likely suboptimal for this cell-based assay.

For HRV-16 (Figure 6B); full protection against infections was recorded at RMN at a concentration of 125 µg/mL as revealed by cell viability percentage (104%) that is similar to untreated cells and much higher than the viability percentage (36%) recorded for HeLa cells treated with the same RMN concentration, 125 µg/mL (Figure 5A). This effect can be attributed to the effective interaction of RMN with HRV-16 resulting in a complete inactivation of the virus. Thus, sufficient RMN interacted with the virus capsid leaving no RMN or HRV to cause cytotoxic effects on HeLa cells. This is consistent with prior literature [5] where RMN at a concentration of 20, 78, 312 µg/mL were reported to entrap or interact with SARS-CoV-2 (an enveloped virus) resulting in virus inactivation. Similarly, rhamnolipids mixtures were also reported to inactivate a non-enveloped virus (poliovirus PV-1) by 40% at a concentration of 50 µg/mL [29]. The obtained data suggested a very narrow window of RMN antiviral activity in this particular HRV assay and also indicated a complete inactivation of HRV-16 (major group) with corresponding inefficient or partial inactivation of HRV-IB (minor group). Interestingly, these findings match with the results obtained from the docking study where more favorable and stable interactions were observed with RMN to major group HRV.

Further experiments were performed to ascertain the results at a narrower concentration range of RMN; 60 to 200 µg/mL. Again, relatively higher cell viability percentages were recorded for both HRV-1B and HRV-16 at an RMN concentration of 125 µg/mL (Figure 6C,D), however, the cell viability percentage was around 60% for both viruses. There was also a wider range of RMN protection observed in this assay, 90–150 μg/mL RMN for HRV-1A (Figure 6C) and 70–150 μg/mL RMN for HRV-16 (Figure 6D), suggesting improved optimization. However incomplete cell viability percentage and large standard error bars seen were correlated to the mechanical loss of cells during assay processing. This was likely responsible for the high variability of data obtained alongside the potential toxic effect of PBS on cells as previously discussed. Therefore, the antiviral activity of RMN against both viruses was re-investigated using DMEM2%FBS to disperse the molecules prior to incubations with HRV.

The antiviral activity of RMN in DMEM containing 2%FBS was tested against HRV-1B and HRV-16 at a broader concentration range (60 to 400 µg/mL) than what was used in prior experiments using PBS (60–200 µg/mL). The obtained data (Figure 7) revealed that the best protection against RV infection was recorded at a higher RMN concentration (200 µg/mL) for both HRV-1B and HRV-16 compared to that recorded for PBS (125 µg/mL). RMN concentrations directly on either side of 200 μg/mL also demonstrated significant protection with both minor and major group RV. RMN concentrations < 150 μg/mL and the highest RMN concentrations did not provide significant protection. Inefficient or partial protection was recorded for HRV-1B with a maximum cell viability percentage of 67% whereas complete protection was recorded for RV-A16 with values > 100% at several RMN concentrations. Complete protection of 100% cell viability for both major and minor group RVs was noted when using neutralizing antiserum human anti-HRV-16 that cross-reacts with HRV-1B [49,50].

Virus protection capabilities over the RMN concentrations tested showed a similar trend to that seen when using RMN dispersed in PBS but with less variability and reduced SDM. Furthermore, a broader range of RMN concentration from 60 to 400 µg/mL was associated with protection against HRV infection, likely due to interactions with viral capsid. Taken collectively, the results obtained with RMN dispersed in PBS and DMEM suggested that RMN might interact more effectively with the HRV-16 capsid (major group) than that of HRV-1B (minor group), resulting in complete and efficient virus inactivation for major group RV, consistent with results obtained from the docking studies. Moreover, it is worth noting that the cell entry mechanism of HRV-1B (via LDLR) [51] differs from HRV-16 (via ICAM-1 receptors) [35] and the corresponding interaction sites with the RV capsids also differ, potentially explaining the differential effect of RMN on major and minor group RV infectivity. Thus, whilst RMN appears to interact with the RV capsid canyon and can therefore easily interfere with major group interaction with ICAM-1, there is a reduced possibility of affecting minor group RV interaction with LDLR as the capsid binding site at the 5-fold axis of symmetry is far removed.

## 4. Conclusions

In this study, rhamnolipids are shown by in silico docking studies to interact with capsid sites called the canyon, found on both major and minor group RVs. Rhamnolipids were shown to bind with higher stability to major group RVs than minor group RVs. We previously demonstrated the superior antimicrobial activity of rhamnolipids nano-micelles (RMN) against multi-drug-resistant bacteria and SARS-CoV-2. Furthermore, RMN was demonstrated to be bio-safe. In the current study, we prepared and characterized RMN, followed by assessing its antiviral activity against major group (RV-16) and minor group (RV-1B) RVs. The results obtained were consistent with the results of in-silico docking studies whereby we found that RMN interfered with RV-16 infection of cells but demonstrated reduced protection against RV-1B infection. This difference can be explained by RMN interacting with the canyon of the RV capsid. It is likely that RMN sterically hinders the engagement of major group RVs with the ICAM-1 entry receptor since major group RVs interact with entry receptors (ICAM-1) that penetrate deep into the canyon. However, since minor group RVs interact with entry receptors (VLDLR) at a site distant from the canyon known as the VP1 5-fold axis of symmetry, RMN binding is less likely to sterically hinder VLDLR engagement and therefore display reduced inhibition of infection in vitro. Partial RMN activity at higher concentrations against minor group RVs can be explained through the stabilization of capsid structure that leads to less efficient uncoating. Further biochemical studies to determine the precise locations of RMN binding sites to RV capsid will better define the potential for RMN to be applied as universal antiviral agents for RV infections. Of course, the challenge of in vivo RMN application remains because to be a useful antiviral, RMN must be present prior to virus exposure. Therefore, RMN may find the best utility as components of hand sanitizer or as surface cleaning agents to reduce the incidence of RV transmission directly or indirectly via fomites.

## Figures and Tables

**Figure 1 cimb-47-00333-f001:**
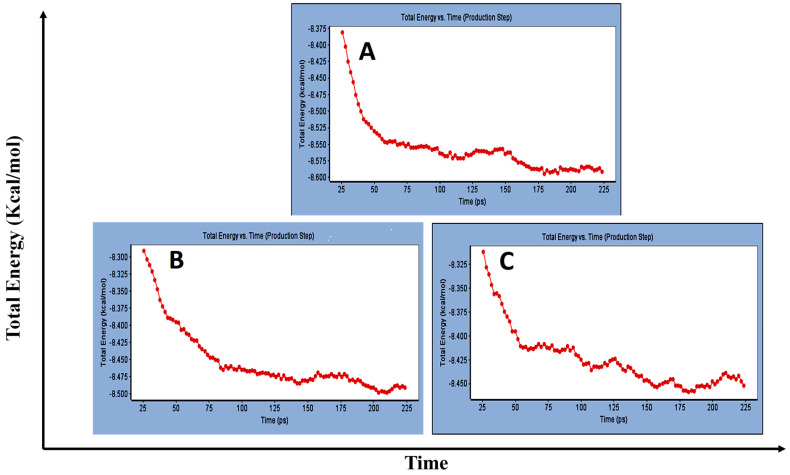
Total energy Vs time in production step during interaction of; (**A**) free protein structure of HRV-14, (**B**) mono-rhamnolipids best conformation pose against free protein structure of HRV-14, (**C**) di-rhamnolipids best conformation pose against free protein structure of HRV-14.

**Figure 2 cimb-47-00333-f002:**
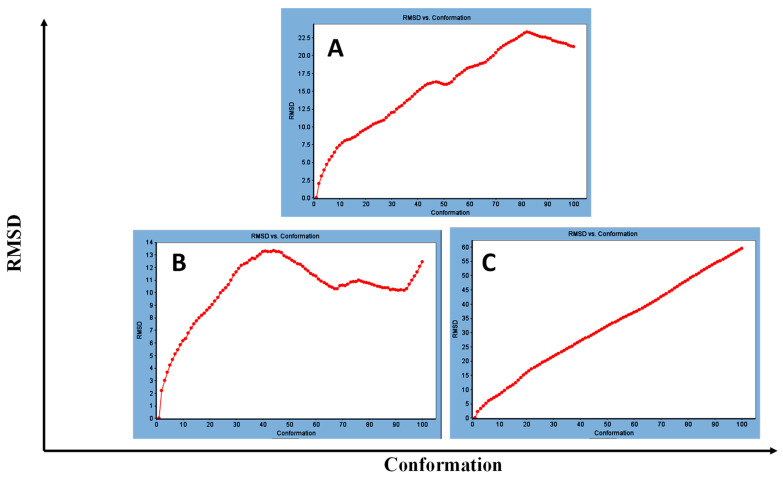
RMSD for (**A**) free protein of HRV-14, (**B**) mono-rhamnolipids best conformation pose against free protein structure of HRV-14, and (**C**) di-rhamnolipids rhamnolipids best conformation pose against free protein structure of HRV-14.

**Figure 3 cimb-47-00333-f003:**
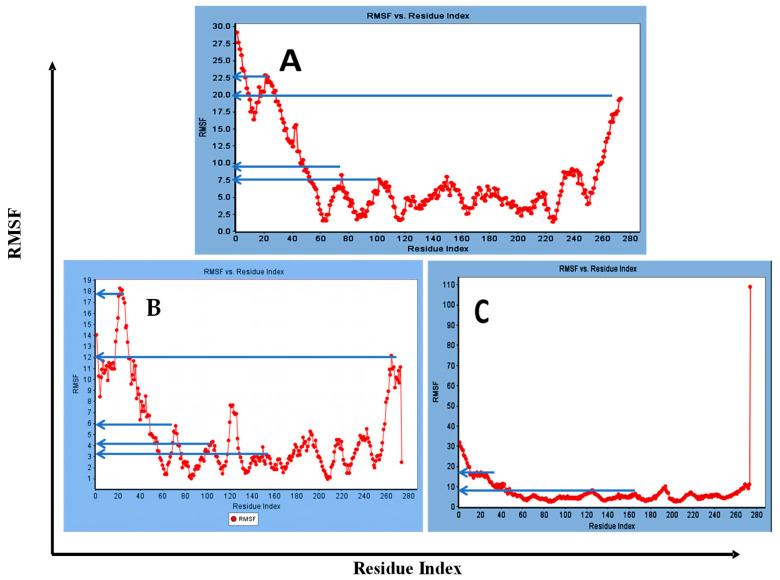
RMSF for (**A**) free protein of HRV-14, (**B**) mono-rhamnolipids best conformation pose against free protein structure of HRV-14, and (**C**) di-rhamnolipids rhamnolipids best conformation pose against free protein structure of HRV-14.

**Figure 4 cimb-47-00333-f004:**
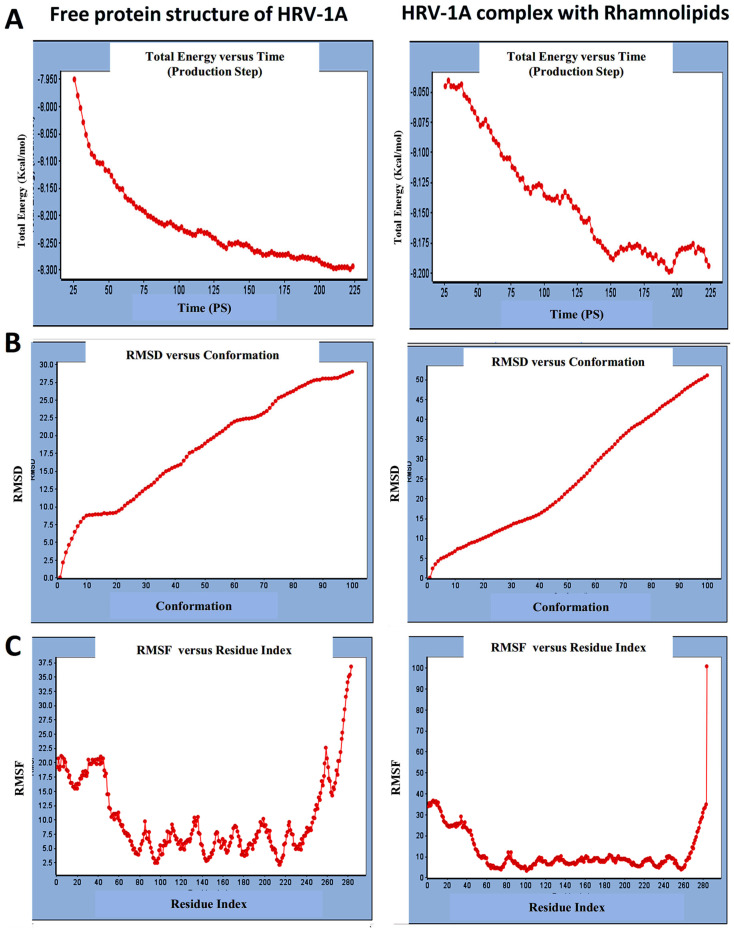
(**A**) Total energy Vs time, (**B**) RMSD, and (**C**) RMSF for free protein structure of HRV-1A and rhamnolipids-HRV-1A complex best conformation pose against HRV-1A free protein structure.

**Figure 5 cimb-47-00333-f005:**
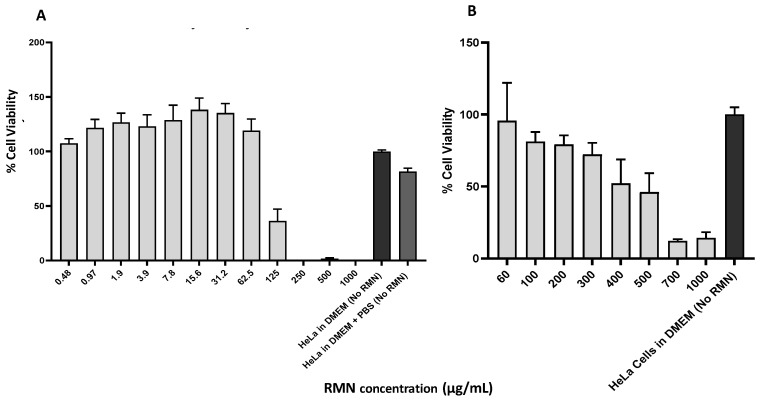
Cytotoxicity of rhamnolipids nano-micelles (RMN) dispersed in (**A**) PBS (10 mM, pH 7.4) and in (**B**) DMEM (2%FBS) on Hela Cells. Results are the average of three independent experiments, with five replicates in each. Error Bars represent standard deviation.

**Figure 6 cimb-47-00333-f006:**
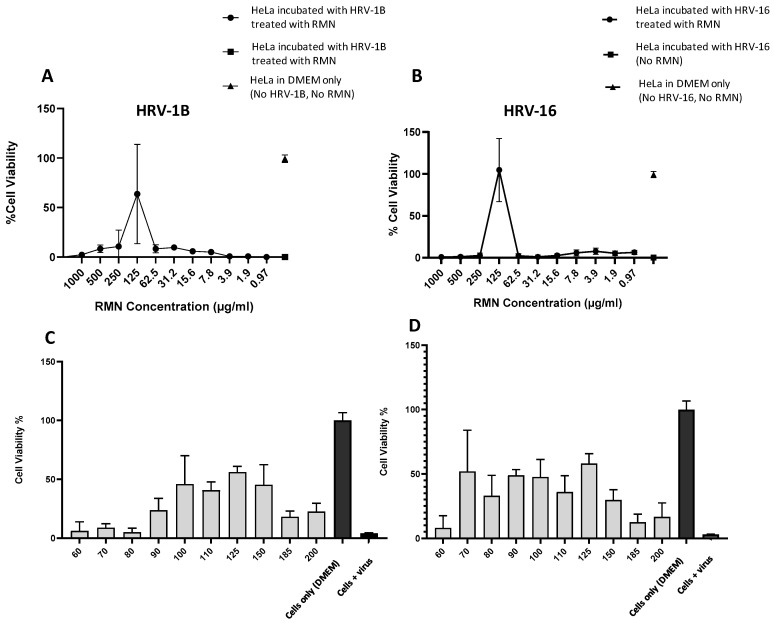
Antiviral activity (virus neutralization) of rhamnolipids nano-micelles (RMN) dispersed in PBS against (**A**) HRV-1B and (**B**) HRV-16 at RMN concentrations from 0.97 to 1000 µg/mL. Results are the average of three independent experiments, with eight replicates in each. Error Bars represent standard deviation. Antiviral activity of RMN dispersed in PBS against (**C**) HRV-1B, and (**D**) HRV-16 at RMN concentration ranged from 60 to 200 was also performed. Results are the average of two independent experiments, with eight replicates in each. Error Bars represent standard deviation.

**Figure 7 cimb-47-00333-f007:**
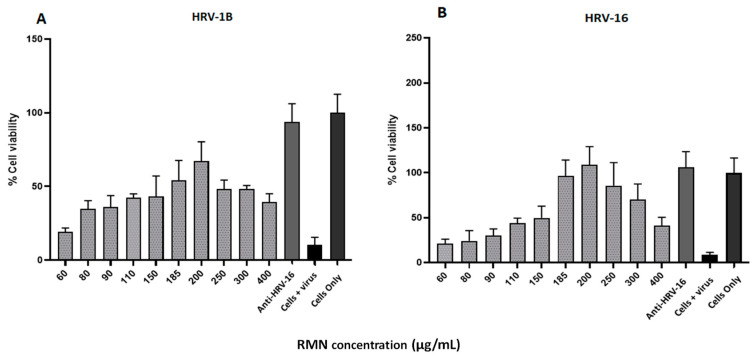
Antiviral activity (virus neutralization) of rhamnolipids nano-micelles (RMN) dispersed in DMEM against HRV-1B virus (**A**) and HRV-16 (**B**) at RMN concentrations ranged from 60 to 400 µg/mL. Results are the average of three independent experiments, with eight replicates for each RMN concentration. Error Bars represent standard deviation.

**Table 1 cimb-47-00333-t001:** Comparison of parameters between RV serotypes used in this study.

RV Serotype	Type	Group	% Identity (Polyprotein)	% Identity (Canyon Floor)	Capsid Structure PDB Code
1A	A	Minor	96	99	1R1A
1B	A				
14	B	Major	<60	51	1R09
16	A				

**Table 2 cimb-47-00333-t002:** C-Docker interaction energy of the rhamnolipids and the antiviral compound, R 61837 with VP1 residues located at the canyon of HRV-14.

Name	Binding Mode	C-Docker Interaction Energy (Kcal/mol)	Key Amino Acids Interactions
Mono-rhamnolipids	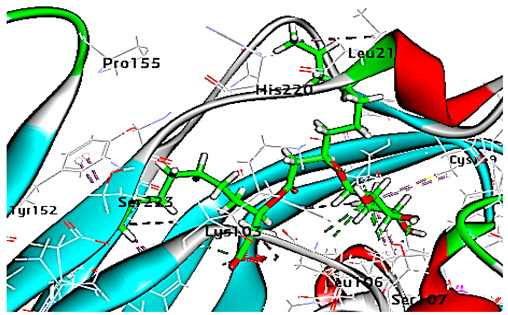	−71.25	1 HBA with Lys103 2 HBA with Leu106 1 HBD with Ser107 Hydrophobic interaction with Leu218, Cys199, Ser223
Di-rhamnolipids	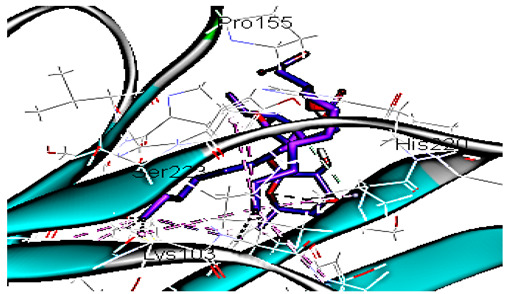	−70.11	1 HBA with His220 Hydrophobic interaction with Lys103, Pro155, Ser223
R 61837	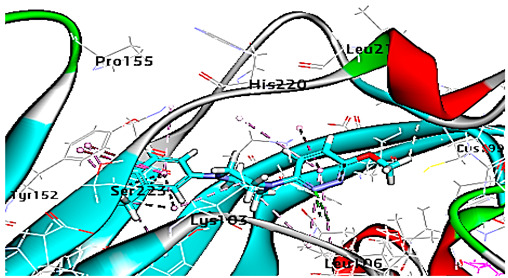	−55.95	1 HBA with Lys103 Hydrophobic interactions with Leu106, Tyr152, Ser223, His220
Mono-rhamnolipids (after MD simulation)	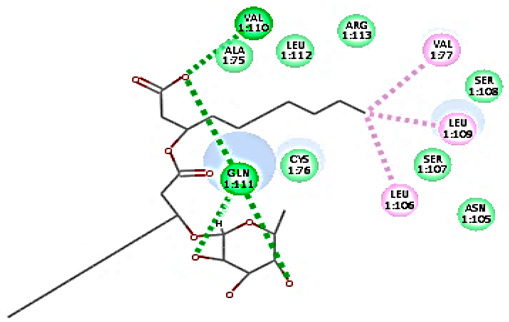	74.66	2 HBD with Val110 and Gln111 2 HBA with Gln111 Hydrophobic interaction with Leu109, Leu106, Val177
Di-rhamnolipids (after MD simulation)	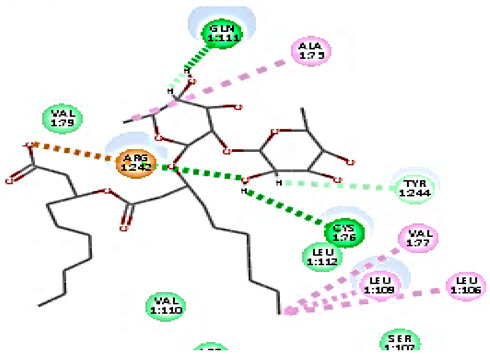	75.40	1 HBA with Arg242 2 HBD with Gln11, Cys176 Hydrophobic interaction with Leu106, Leu109, Val177, Ala175, Arg242
MD = Molecular dynamic			

**Table 3 cimb-47-00333-t003:** C-Docker interaction energy of the rhamnolipids and the antiviral compound, WIN 53338 with VP1 residues located at the canyon of HRV-1A.

Name	Binding Mode	C-Docker Interaction Energy (Kcal/mol)	Key Amino Acids Interactions
Mono-rhamnolipids	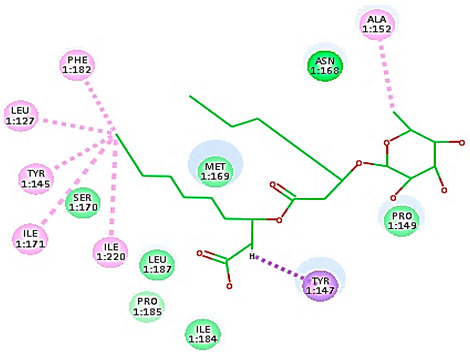	−50.4	Hydrophobic interaction with Leu127, Tyr145, Tyr147,Ala152, Ile171, Phe182, Ile220
Di-rhamnolipids	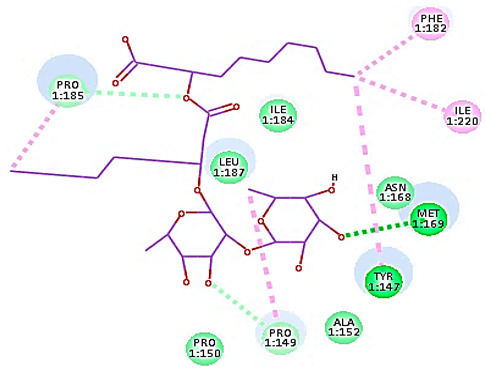	−53.7	1 HBA with Met169 Hydrophobic interaction with Tyr147, Pro149 Ile171, Phe182, Pro185, Ile220
WIN 53338	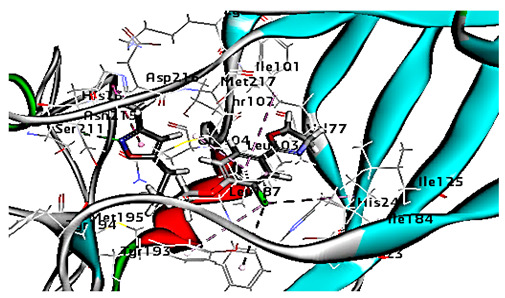	61.25	1 HBA with ASN215 Hydrophobic interaction with Ile101 Leu103 Ile125 Tyr147, PIle171, Pro185, Leu187, Ile220 Ile184 Tyr198 Met217 His268
Mono-rhamnolipids (after MD simulation)	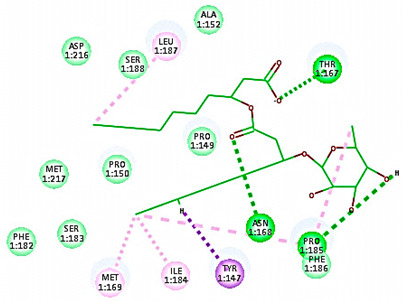	54.36	3 HBA with Thr167, Asp168, Pro185 Hydrophobic interaction with Tyr147, Ile184, Met169, Pro185, Ile187
Di-rhamnolipids (after MD simulation)	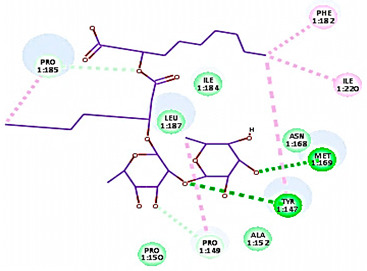	55.93	4 HBA with Met169, Tyr147, 185, Hydrophobic interaction with Tyr147, Pro149 Ile171, Phe182, Pro185, Ile220,
MD = Molecular dynamic			

## Data Availability

Reasonable requests for data are available by contacting either of the corresponding authors.

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
