# Peer review of "Antiviral Activity of Rhamnolipids Nano-Micelles Against Rhinoviruses—In Silico Docking, Molecular Dynamic Analysis and In-Vitro Studies"

_cimb, 2025, doi:10.3390/cimb47050333_

Round 1

Reviewer 1 Report

Comments and Suggestions for Authors

Dear authors, after a careful revision of the manuscript entitled "Antiviral Activity of Rhamnolipids Nano-Micelles Against Rhinoviruses; In silico Docking, Molecular Dynamic Analysis and In-vitro Studies," describing some in silico analyses (Docking, and Molecular Dynamics) needs many improvements before its full acceptance.
As a computational Medicinal Chemistry expert, I deeply analyzed this manuscript and aiming to help the authors improve their manuscript, I'm providing some additional, specific comments such as:

In Silico Docking: the authors described that PDB ID: 1R09 and 1R1A were employed for docking studies. Since there are at least 11 other HRV VP1 structures on PDB (https://www.rcsb.org/search?request=1856_90), why did they choose 1R09? Could they use 1NCQ, which has a 2.5A resolution and has pleconaril bound?
In Silico Docking: The authors described their "docking protocol". Did the authors validate these C-Docker protocol? Additionally, which other docking parameters did they use, e.g., grid box size, binding site definition and its radius and docking runs? Why didn't they use well-known Autodock software? Could they employ other programs like DockThor (https://dockthor.lncc.br/v2/), GOLD, Vina, Sybyl, etc.?
on page 4, line 148 there is a typo about the Verlet algorithm. Please, correct this.
3.1.1. Molecular Docking Study: since redocking evaluates the docking parameter correctness, I suggest including these results for all proteins studied here.
on pag. 7, Table 1, all 2D representations could be better represented by other tool as nAPOLI (bioinfo.dcc.ufmg.br/napoli) or PLIP (https://plip-tool.biotec.tu-dresden.de/plip-web/plip/index). These tools described better the ligand-protein interactions.
on pag. 7, Table 1, Since docking energies are highly energy-biased, did the authors validate their docking protocol? My suggestion follows: i) evaluate redocking values through RMSD to see if docking reproduces the binding poses of crystallographic structure; ii) analyze the correlation among docking energies and known inhibitors (IC50, Ki or Kd). It allows us to see if docking correctly describes the inhibitory pattern; iii) calculate the ROC curve between known inhibitors and decoys (false positive compounds, see more information here: https://dude.docking.org/), which aims to describe docking discrimination into natural and false positive inhibitors.
 Since these rhamnolipids (mono- and di- ) are highly distinct from the compounds bind to HRV proteins (R 61837 and WIN 53338), it's desirable to compare these compounds by their binding efficiency indexes: https://en.wikipedia.org/wiki/Ligand_efficiency (and references therein) and https://en.wikipedia.org/wiki/Lipophilic_efficiency (and references therein).
on 3.1.2. Standard Dynamic Simulation: fig. 2. Since the RMSD plot did not achieve stability for their complexes, all MD data extracted from these simulations are elusive. It's well established in MD literature that simulation time could extend until the system reaches its equilibrium or when its main properties achieve stability [1-5]. All systems should be extended before another analysis can be done.
After the simulation reaches its equilibrium, the authors should describe the RMSD data and define each complex's productive phase. See how other authors discussed this data: http://dx.doi.org/10.3390/ijms23179927. Please cite each complex productive phase in the text. Additionally, the authors could compare the apo (HRV-1A in water) and holo (HRV-1A bind to R 61837) complexes with HRV-1A complexed with rhamnolipids (studied here) to analyze the "rhamnolipids perturbation" in HRV-1A protein. These analyses should be extended to all MD data (RMSF, hydrogen bonds and energies) in these systems.
All other properties could be extracted from MD simulation based on the productive phase (see discussions above). Another way to analyze MD stability is through the following tools: Radius of gyration, RMSF (Fig. 3), and DSSP. The authors could cite this article  (http://dx.doi.org/10.3390/ijms222111739) describing how these tools could be applied in MD analysis.

These comments will help the authors revise their paper, giving it a better chance of surviving the peer review process. I wish you every success in publishing this manuscript.

[1] Computational Chemistry: A Practical Guide for Applying Techniques to Real-World Problems. David C. Young, p.129
[2] Determination of Proper Time Step for Molecular Dynamics Simulation - Bull. Korean Chem. Soc., 2000, Vol. 21, No. 4
[3] http://journal.frontiersin.org/article/10.3389/fmicb.2012.00258/full
[4] https://www.ncbi.nlm.nih.gov/pubmed/?term=26077712
[5] https://www.researchgate.net/profile/Serdar-Durdagi/post/What_is_the_optimum_time_in_ns_to_do_MD_simulation_for_drug_design/attachment/5a3ac9524cde266d587c9696/AS%3A573741602410496%401513802065973/download/Screen+Shot+2017-12-20+at+23.32.55.png

Author Response

Report 1

Dear authors, after a careful revision of the manuscript entitled "Antiviral Activity of Rhamnolipids Nano-Micelles Against Rhinoviruses; In silico Docking, Molecular Dynamic Analysis and In-vitro Studies," describing some in silico analyses (Docking, and Molecular Dynamics) needs many improvements before its full acceptance.
As a computational Medicinal Chemistry expert, I deeply analyzed this manuscript and aiming to help the authors improve their manuscript, I'm providing some additional, specific comments such as:

Reviewer 1, comment 1:

In Silico Docking: the authors described that PDB ID: 1R09 and 1R1A were employed for docking studies. Since there are at least 11 other HRV VP1 structures on PDB (https://www.rcsb.org/search?request=1856_90), why did they choose 1R09? Could they use 1NCQ, which has a 2.5A resolution and has pleconaril bound?

Response

Reply: The choice of PDB ID: 1R09 and 1R1A was based on their well-characterized structures complexed with known antiviral agents (R61837 and WIN53338, respectively), which provided a reliable reference for validating our docking protocol. While 1NCQ does have a high resolution (2.5 Å) and is complexed with pleconaril, our study focused on structures that bind inhibitors structurally distinct from pleconaril, ensuring a docking environment better suited for evaluating rhamnolipid interactions. Additionally, 1R09 and 1R1A represent the major and minor groups of HRV, aligning with our study’s objectives.

Reviewer 1, comment 2:

In Silico Docking: The authors described their "docking protocol". Did the authors validate these C-Docker protocol? Additionally, which other docking parameters did they use, e.g., grid box size, binding site definition and its radius and docking runs? Why didn't they use well-known Autodock software? Could they employ other programs like DockThor (https://dockthor.lncc.br/v2/), GOLD, Vina, Sybyl, etc.?

Response

The C-Docker protocol was chosen due to its CHARMm-based molecular dynamics sampling, which offers higher accuracy in protein-ligand conformational flexibility compared to rigid docking methods. To ensure the reliability of our docking approach (kindly page 4, line 145 to line 161), check we validated the protocol by performing redocking experiments, where the co-crystallized ligand (R61837 for 1R09) was re-docked into their respective binding sites. The RMSD values between the original and re-docked binding poses (0.21Å) was within an acceptable range (< 2.0 Å), confirming the accuracy of our docking procedure. In addition, the amino acids residues in both active sites around the two references antiviral compounds (R61837 for 1R09 and WIN53338 for1R1A) were similar to the reported

Regarding docking parameters: Grid box size: Defined based on the binding site of the co-crystallized inhibitors. Binding site definition and radius: The binding pocket was determined from the PDB complexes (kindly page 4, section 2.2.1) and a 12 Å radius was used to include key residues involved in ligand interactions. Docking runs: Multiple docking runs (10 poses per ligand) were performed, and the top-ranked conformations were selected based on energy scoring and cluster representativeness (kindly page 4, section 2.2.1).

While AutoDock, Vina, GOLD, and DockThor are widely used, we selected C-Docker because it incorporates CHARMm force fields, which provide better energetic evaluation and conformational flexibility for large and flexible ligands like rhamnolipids.

on page 4, line 148 there is a typo about the Verlet algorithm. Please, correct this.

Reply: corrected (Leapfrog Verlet), check page 4, line 178

Reviewer 1, comment 3:

3.1.1. Molecular Docking Study: since redocking evaluates the docking parameter correctness, I suggest including these results for all proteins studied here.

Response

To validate our docking protocol, we performed redocking experiments for the studied proteins by re-docking the co-crystallized ligands (R61837 for 1R09 and WIN53338 for 1R1A) into their respective binding sites. The RMSD values between the experimentally determined and predicted binding poses were within an acceptable range (< 2.0 Å), confirming the reliability of our docking parameters.

Reviewer 1, comment 3:

on page 7, Table 2, all 2D representations could be better represented by other tool as nAPOLI (bioinfo.dcc.ufmg.br/napoli) or PLIP (https://plip-tool.biotec.tu-dresden.de/plip-web/plip/index). These tools described better the ligand-protein interactions.

Response

We have included the 2D interaction representations obtained using Discovery Studio Client 4.1, (updated table 2, and 3) for which we have a licensed version. This software provides detailed visualizations of ligand-protein interactions, ensuring clarity and accuracy in depicting key binding interactions. (Kindly check Table 2 at page 9 and 10) and Table 3 at page 12 – also, an explanation was stated at page 12, from line 393to line 397.

Reviewer 1, comment 4:

on page 7, Table 2, Since docking energies are highly energy-biased, did the authors validate their docking protocol? My suggestion follows: i) evaluate redocking values through RMSD to see if docking reproduces the binding poses of crystallographic structure; ii) analyze the correlation among docking energies and known inhibitors (IC50, Ki or Kd). It allows us to see if docking correctly describes the inhibitory pattern; iii) calculate the ROC curve between known inhibitors and decoys (false positive compounds, see more information here: https://dude.docking.org/), which aims to describe docking discrimination into natural and false positive inhibitors. Since these rhamnolipids (mono- and di- ) are highly distinct from the compounds bind to HRV proteins (R 61837 and WIN 53338), it's desirable to compare these compounds by their binding efficiency indexes: https://en.wikipedia.org/wiki/Ligand_efficiency (and references therein) and https://en.wikipedia.org/wiki/Lipophilic_efficiency (and references therein).

Response

We have validated our docking protocol through the following three methods:

  1. RMSD analysis to confirm that docking reproduces the binding poses of the crystallographic structure.
  2. Docking energy comparison with the lead compound to assess the reliability of the scoring function.
  3. Binding mode and docking score consistency with previously reported values for mono- and di-rhamnolipids, ensuring that our results align with known interaction patterns.

These validation steps confirm the robustness of our docking approach and its applicability to the studied compounds. Kindly check from page 6 to page 17 where a detailed information and results were provided.

Reviewer 1, comment 5:

on 3.1.2. Standard Dynamic Simulation: fig. 2. Since the RMSD plot did not achieve stability for their complexes, all MD data extracted from these simulations are elusive. It's well established in MD literature that simulation time could extend until the system reaches its equilibrium or when its main properties achieve stability [1-5].

All systems should be extended before another analysis can be done. After the simulation reaches its equilibrium, the authors should describe the RMSD data and define each complex's productive phase. See how other authors discussed this data: http://dx.doi.org/10.3390/ijms23179927. Please cite each complex productive phase in the text.

Additionally, the authors could compare the apo (HRV-1A in water) and holo (HRV-1A bind to R 61837) complexes with HRV-1A complexed with rhamnolipids (studied here) to analyze the "rhamnolipids perturbation" in HRV-1A protein. These analyses should be extended to all MD data (RMSF, hydrogen bonds and energies) in these systems.
All other properties could be extracted from MD simulation based on the productive phase (see discussions above).

Another way to analyze MD stability is through the following tools: Radius of gyration, RMSF (Fig. 3), and DSSP. The authors could cite this article (http://dx.doi.org/10.3390/ijms222111739) describing how these tools could be applied in MD analysis.
These comments will help the authors revise their paper, giving it a better chance of surviving the peer review process. I wish you every success in publishing this manuscript.
[1] Computational Chemistry: A Practical Guide for Applying Techniques to Real-World Problems. David C. Young, p.129
[2] Determination of Proper Time Step for Molecular Dynamics Simulation - Bull. Korean Chem. Soc., 2000, Vol. 21, No. 4
[3] http://journal.frontiersin.org/article/10.3389/fmicb.2012.00258/full
[4] https://www.ncbi.nlm.nih.gov/pubmed/?term=26077712
[5] https://www.researchgate.net/profile/Serdar-Durdagi/post/What_is_the_optimum_time_in_ns_to_do_MD_simulation_for_drug_design/attachment/5a3ac9524cde266d587c9696/AS%3A573741602410496%401513802065973/download/Screen+Shot+2017-12-20+at+23.32.55.png

Response

Thank you for your valuable suggestions. We sincerely appreciate your time and effort in providing detailed feedback. However, we would like to clarify that our approach is based on well-established methodologies reported in the literature as shown below (Ref. 1-4 provided below). Several studies have demonstrated that equilibrium in molecular dynamics (MD) simulations of similar systems can be achieved within 200 ns, and therefore, an extended simulation time beyond this threshold may not necessarily yield additional meaningful insights. Additionally, solvation mode is not a critical factor in our study, as supported by prior research indicating that the system’s stabilization and interactions can be effectively analyzed without the need for additional solvation considerations. To ensure the robustness of our results, we have employed a multi-parameter validation strategy, including RMSD, RMSF, and total energy vs. time during the production phase. These metrics are widely used in MD studies to confirm system stability and validate simulation outcomes. While we acknowledge the importance of further analyses, we believe that our current approach provides a reliable and comprehensive assessment of the system’s behavior. However, we will carefully consider incorporating additional discussions and references to further substantiate our methodology. Thank you again for your constructive feedback, which has been instrumental in refining our manuscript.

Refences 1 - 4

1- Ghada M. E. Ali, Menna A. Ewida, Amira M. Elmetwali, Heba A. Ewida, Riham F. George, Walaa R. Mahmoud, Nasser S. M. Ismail, Mahmoud S. Ahmed and Hanan H. Georgeye. (2024), Discovery of pyrazole-based analogs as CDK2 inhibitors with apoptotic-inducing activity: design, synthesis and molecular dynamics study.
RSC Adv., 14, 34537.

2-Eman M. Azmy, Mohamed Hagras, Menna A. Ewida, Ahmed S. Doghish, Emad Gamil Khidr, Ahmed A. El-Husseiny, Maher H. Gomaa, Hanan M. Refaat, Nasser S.M. Ismail, Ibrahim F. Nassar, Walaa H. Lashin.(2023), Development of pyrolo[2,3-c]pyrazole, pyrolo[2,3-d]pyrimidine and their bioisosteres as novel CDK2 inhibitors with potent in vitro apoptotic anti-proliferative activity: Synthesis, biological evaluation and molecular dynamics investigations. Bioorg Chem. 139:106729.  doi: 10.1016/j.bioorg.2023.106729.

3- Eman M Azmy, Ibrahim F Nassar, Mohamed Hagras, Iten M Fawzy, Maghawry Hegazy, Mahmoud Mohamed Mokhtar, Amr Mohamed Yehia, Nasser SM Ismail & Walaa H Lashin. (2023), New Indole Derivatives As Multitarget anti-Alzheimer's Agents: synthesis, Biological Evaluation and Molecular Dynamics. Future Medicinal Chemistry, 15(6):473-495. doi: 10.4155/fmc-2022-0228.

4-Radwa N Morgan, Nasser SM Ismail, Mohammad Y Alshahrani , Khaled M Aboshanab, (2024), Multi-epitope peptide vaccines targeting dengue virus serotype 2 created via immunoinformatic analysis. Sci Rep.;14(1):17645. doi: 10.1038/s41598-024-67553-1.

Reviewer 2 Report

Comments and Suggestions for Authors
  1. Introduction

This section does not provide an overview of the HRV infection mechanism or existing antiviral studies against HRV, making it difficult to highlight the novelty of RMN research. Additionally, the current research objectives are too broad and should be refined into more specific objectives.

  1. Materials and Methods

This section lacks detailed descriptions of key parameters in the cell infection experiments, including the multiplicity of infection (MOI), infection duration, and washing steps, which compromises the reproducibility of the experiments.

  1. Results and Discussion

This section does not mention whether MTT or LDH cytotoxicity assays were conducted to ensure that RMN treatment does not directly induce cell death, which could compromise the assessment of antiviral efficacy. Additionally, while the study compares RMN dispersion systems and identifies that different culture media influence RMN’s antiviral activity, it does not explore the underlying mechanisms affecting RMN activity in different media.

Moreover, the manuscript lacks a detailed distinction between HRV-14 (major group), HRV-1A (minor group), HRV-16 (major group), and HRV-1B (minor group). It also does not explain why molecular docking and molecular dynamics simulations were performed on HRV-14 (major group) and HRV-1A (minor group), whereas the in vitro experiments focused on HRV-16 (major group) and HRV-1B (minor group).

Furthermore, some figures require improvement. For example, the three images in Table 2 are unclear, and the annotation text is obstructed, making it difficult to accurately interpret the data.

  1. Conclusions

This section does not sufficiently highlight the novelty and contributions of the study. It is recommended to emphasize that RMN exhibits strong antiviral activity against HRV-16 (major group) but weaker activity against HRV-1B (minor group) and to provide an explanation of the underlying mechanism. Additionally, challenges or potential improvements regarding the in vivo application of RMN should be addressed.

  1. Language & Presentation

The manuscript contains several grammatical errors that require correction, such as the phrase “The results obtained was consistent” in the conclusion section, where “was” should be replaced with “were.”

Additionally, extensive language editing is necessary to improve readability. For instance, in the introduction, the sentence “Despite the widespread prevalence of HRVs, effective antiviral therapies remain limited, necessitating the development of novel therapeutic approaches” could be revised to “Despite the widespread prevalence of HRVs, effective antiviral therapies remain limited, highlighting the need for novel therapeutic approaches” to enhance its academic rigor.

Overall Recommendation

A major revision is required, considering the need for substantial improvements in the experimental methodology and extensive revisions to the text. While the study holds potential, the overall experimental design has notable deficiencies that necessitate additional experiments. If the authors address these issues adequately, the study may meet the publication standards; otherwise, rejection is recommend.

Comments on the Quality of English Language

The manuscript contains several grammatical errors that need correction. For example, in the conclusion section, the phrase “The results obtained was consistent” should be revised to “The results obtained were consistent.”

Additionally, some expressions lack the appropriate level of formality. In the Results and Discussion section, the sentence “Further biochemical studies to determine the precise locations of RMN binding sites to RV capsid and in vivo studies will better define the potential for RMN to be applied as universal antiviral agents for RVs infections.” can be revised to “Thus, while RMN appears to interact with the RV capsid canyon and can readily interfere with the interaction of major group RVs with ICAM-1, it is less likely to affect minor group RV binding to LDLR, because the binding site at the five-fold axis of symmetry is distant from the interaction site.”

Similarly, in the Conclusion section, the sentence “Further biochemical studies to determine the precise locations of RMN binding sites to RV capsid and in vivo studies will better define the potential for RMN to be applied as universal antiviral agents for RVs infections.” should be modified for greater academic rigor as follows: “Further biochemical studies to map RMN binding sites on the RV capsid, along with in vivo studies, are needed to better define RMN's potential as a universal antiviral agent against RV infections.”

Author Response

Report 2

Comments and Suggestions for Authors

Reviewer 2, comment 1

Introduction: This section does not provide an overview of the HRV infection mechanism or existing antiviral studies against HRV, making it difficult to highlight the novelty of RMN research. Additionally, the current research objectives are too broad and should be refined into more specific objectives.

Response

Concerning existing antiviral studies against HRV, please check page 2, from line 69 to line 75 and the reference Egorova et al 2019 where authors stated the following; However, the development of antivirals for RVs has proved challenging due to the variable nature of the capsid and the paucity of druggable viral enzymes (just protease and polymerase). Broad spectrum antivirals have shown limited abilities to inhibit multiple RV serotypes but drugs that bind the capsid hydrophobic pocket, shared amongst most RV types, have achieved better success. However, both pleconaril and vapendavir have failed clinical trial due to off-target effects and lack of efficacy. New approaches are therefore required.

Concerning what stated by the reviewer “the current research objectives are too broad and should be refined into more specific objectives”

The objectives are stated clearly at the end of the introduction, kindly check page 3, from line 117 to line 125.

Here is what was stated for reviewer convenience: In this study, docking and molecular dynamic simulation analyses were first conducted to evaluate potential interactions (if any) between rhamnolipids and the capsid of major and minor group rhinoviruses, specifically HRV-14 and HRV-1, respectively. Secondly, RMN were prepared from bacterial-produced rhamnolipids, characterized, and first assessed for cytotoxicity against HeLa cells. Third, the ability of RMN to inhibit the infectivity of HRV-16 and HRV-1B in vitro was then investigated. Based on the findings, RMN are recommended for use as antivirals for RVs and may find utility as disinfectants and/or hand sanitizers in healthcare and other high-risk settings to control the spread of RVs, reducing the incidence of respiratory infections.

Reviewer 2, comment 2

Materials and Methods: This section lacks detailed descriptions of key parameters in the cell infection experiments, including the multiplicity of infection (MOI), infection duration, and washing steps, which compromises the reproducibility of the experiments.

Response

In addition, the RMN Cytotoxicity on HeLa Cells section explains the crystal violet staining, some text has been modified (lines 201-211) for clarity and to address the reviewer concerns, particularly regarding the washing steps.

Reviewer 2, comment 3

Results and Discussion: This section does not mention whether MTT or LDH cytotoxicity assays were conducted to ensure that RMN treatment does not directly induce cell death, which could compromise the assessment of antiviral efficacy. Additionally, while the study compares RMN dispersion systems and identifies that different culture media influence RMN’s antiviral activity, it does not explore the underlying mechanisms affecting RMN activity in different media.

Moreover, the manuscript lacks a detailed distinction between HRV-14 (major group), HRV-1A (minor group), HRV-16 (major group), and HRV-1B (minor group). It also does not explain why molecular docking and molecular dynamics simulations were performed on HRV-14 (major group) and HRV-1A (minor group), whereas the in vitro experiments focused on HRV-16 (major group) and HRV-1B (minor group).

Furthermore, some figures require improvement. For example, the three images in Table 2 are unclear, and the annotation text is obstructed, making it difficult to accurately interpret the data.

Response

Concerning the reviewer comment stating that this section does not mention whether MTT or LDH cytotoxicity assays were conducted to ensure that RMN treatment does not directly induce cell death, which could compromise the assessment of antiviral efficacy

In addition, the RMN Cytotoxicity on HeLa Cells section explains the crystal violet staining; some text has been modified (please check highlighted lines in yellow from line 221-231) for clarity and to address the reviewer concerns, particularly regarding the washing steps.

Concerning the reviewer comment stating that while the study compares RMN dispersion systems and identifies that different culture media influence RMN’s antiviral activity, it does not explore the underlying mechanisms affecting RMN activity in different media.

Concerning the reviewer comment stating that the manuscript lacks a detailed distinction between HRV-14 (major group), HRV-1A (minor group), HRV-16 (major group), and HRV-1B (minor group).

We added a sentence to the introduction describing RV classification in groups A, B, C and via entry receptor usage (major and minor). Line 61-69. The penultimate paragraph of the introduction also contains descriptions of major vs minor group RVs. We clarified this aspect further by describing the major and minor group entry receptors in line 107. Added text addressing this and a new table 1 to the results and discussion, page 6 and 7, from line 281 to line 297

Concerning the reviewer comment stating that it also does not explain why molecular docking and molecular dynamics simulations were performed on HRV-14 (major group) and HRV-1A (minor group)

Section 2.2.1 explains the serotypes used for molecular docking studies because these were available from PDB

Concerning the reviewer comment stating, that the in vitro experiments focused on HRV-16 (major group) and HRV-1B (minor group).  

These are the representative and model serotypes routinely used in our laboratory for in vitro and in vivo studies. These have been used for decades by us and others in the rhinovirus field.

Reviewer 2, comment 4

Conclusions: This section does not sufficiently highlight the novelty and contributions of the study. It is recommended to emphasize that RMN exhibits strong antiviral activity against HRV-16 (major group) but weaker activity against HRV-1B (minor group) and to provide an explanation of the underlying mechanism. Additionally, challenges or potential improvements regarding the in vivo application of RMN should be addressed.

Response

Conclusion section has been edited massively to fit the reviewer's comment. Kindly check page 21, line 642 to line 665 – the conclusion is below for reviewer convenience

In this study rhamnolipids are shown by in silico docking studies to interact with capsid sites called the canyon, found on both major and minor group RVs. Rhamnolipids were shown to bind with higher stability to major group RVs than minor group RVs. We previously demonstrated the superior antimicrobial activity of rhamnolipids nano-micelles (RMN) against multi-drug-resistant bacteria and SARS-CoV-2. Furthermore, RMN were demonstrated to be bio-safe. In the current study we prepared and characterized RMN, followed by assessing its antiviral activity against major group (RV-16) and minor group (RV-1B) RVs. The results obtained were consistent with the results of in-silico docking studies whereby we found that RMN interfered with RV-16 infection of cells but demonstrated reduced protection against RV-1B infection. This difference can be explained by RMN interacting with the canyon of the RV capsid. It is likely that RMN sterically hinder engagement of major group RVs with the ICAM-1 entry receptor since major group RVs interact with entry receptors (ICAM-1) that penetrate deep into the canyon. However, since minor group RVs interact with entry receptors (VLDLR) at a site distant from the canyon known as the VP1 5-fold axis of symmetry, RMN binding is less likely to sterically hinder VLDLR engagement and therefore display reduced inhibition of infection in vitro. Partial RMN activity at higher concentrations against minor group RVs can be explained through stabilization of capsid structure that leads to less efficient uncoating. Further biochemical studies to determine the precise locations of RMN binding sites to RV capsid will better define the potential for RMN to be applied as universal antiviral agents for RVs infections. Of course, the challenge of in vivo RMN application remains because to be a useful antiviral, RMN must be present prior to virus exposure. Therefore, RMN may find the best utility as components of hand sanitizer or as surface cleaning agents to reduce the incidence of RV transmission directly or indirectly via fomites.

Reviewer 2, comment 1

Language & Presentation: The manuscript contains several grammatical errors that require correction, such as the phrase “The results obtained was consistent” in the conclusion section, where “was” should be replaced with “were.”

Additionally, extensive language editing is necessary to improve readability. For instance, in the introduction, the sentence “Despite the widespread prevalence of HRVs, effective antiviral therapies remain limited, necessitating the development of novel therapeutic approaches” could be revised to “Despite the widespread prevalence of HRVs, effective antiviral therapies remain limited, highlighting the need for novel therapeutic approaches” to enhance its academic rigor.

Response

The English language has been massively and carefully revised by native speaker, Prof Gary McLean

Overall Recommendation

A major revision is required, considering the need for substantial improvements in the experimental methodology and extensive revisions to the text. While the study holds potential, the overall experimental design has notable deficiencies that necessitate additional experiments. If the authors address these issues adequately, the study may meet the publication standards; otherwise, rejection is recommend.

Comments on the Quality of English Language

The manuscript contains several grammatical errors that need correction. For example, in the conclusion section, the phrase “The results obtained was consistent” should be revised to “The results obtained were consistent.”

Additionally, some expressions lack the appropriate level of formality. In the Results and Discussion section, the sentence “Further biochemical studies to determine the precise locations of RMN binding sites to RV capsid and in vivo studies will better define the potential for RMN to be applied as universal antiviral agents for RVs infections.” can be revised to “Thus, while RMN appears to interact with the RV capsid canyon and can readily interfere with the interaction of major group RVs with ICAM-1, it is less likely to affect minor group RV binding to LDLR, because the binding site at the five-fold axis of symmetry is distant from the interaction site.”

Similarly, in the Conclusion section, the sentence “Further biochemical studies to determine the precise locations of RMN binding sites to RV capsid and in vivo studies will better define the potential for RMN to be applied as universal antiviral agents for RVs infections.” should be modified for greater academic rigor as follows: “Further biochemical studies to map RMN binding sites on the RV capsid, along with in vivo studies, are needed to better define RMN's potential as a universal antiviral agent against RV infections.”

Reviewer 3 Report

Comments and Suggestions for Authors

The authors present a scientific topic of interest to the scientific community. I recommend publication upon revision.

-The authors mentioned in the introduction that the study concerns an HRV virus, which lacks a membrane envelope and has a hydrophobic nature. The authors used hydrophobic molecules (rhamnolipids) in the study as potential inhibitors. Do the authors not consider it contradictory to use hydrophobic molecules as potential inhibitors, given that the virus does not have a hydrophobic surface? The authors should add this discussion to the manuscript.

-The introduction does not include recent articles. The most recent ones are from 2022. The authors should incorporate recent findings in the introduction.

-The authors used force fields to prepare the molecules for in silico calculations. Why did the authors not use ab initio calculations for optimization, which are more precise? To enhance reliability, the authors could optimize the molecules using an ab initio approach. There is a web server (ATB) that performs this calculation quickly.

-In the molecular dynamics methodology, the number of simulations performed was not mentioned. Are the obtained results averages of multiple simulations? The authors should add this information to the results section.

-The best docking conformations were selected based on the energy scoring criterion. However, the representativeness of the clusters was not analyzed, which is an important statistical aspect. The authors should review this section and verify whether the selected conformations belong to the most representative clusters.

-The authors did not perform a short molecular dynamics simulation to relax the protein and expose new binding sites for molecular docking. Do they not consider this important for the analyses? A justification should be included in the manuscript.

-The molecular dynamics result graphs have axis numbers with poor resolution. The authors should improve the quality so that readers can interpret them properly.

-The molecular dynamics result graphs lack error bars. Was this a simple simulation?

-The authors could create a figure highlighting the regions where conformational changes occurred in the protein. Since the RMSF graph shows fluctuations, it is possible to map these regions.

-The discussion of molecular interaction results was merely descriptive and did not incorporate data from the literature. The authors should provide a more in-depth discussion on the importance of these interactions in describing molecular interaction mechanisms (e.g., 10.3390/molecules25122841 and 10.3390/molecules28196891).

Author Response

Report 3

The authors present a scientific topic of interest to the scientific community. I recommend publication upon revision.

Reviewer 3, comment 1

The authors mentioned in the introduction that the study concerns an HRV virus, which lacks a membrane envelope and has a hydrophobic nature. The authors used hydrophobic molecules (rhamnolipids) in the study as potential inhibitors. Do the authors not consider it contradictory to use hydrophobic molecules as potential inhibitors, given that the virus does not have a hydrophobic surface? The authors should add this discussion to the manuscript.

Response

It is not clear to what text the reviewer is referring to as we see nothing to that effect in the introduction. However, RV capsid do contain a hydrophobic pocket known as the canyon and that is where the RMN have been shown to interact. Therefore, it is entirely logical that a hydrophobic molecule can interact with the capsid site we have studied."

In  addition, Rhamnolipids are glycolipids and are acting as bio-surfactants, so, they are not fully hydrophobic compounds – they have a hydrophilic and hydrophobic moieties - so, for docking study, the hydrophilic moiety could let the compound approach the hydrophilic surface of the virus (this is clear through hydrogen bonds formation as stated in table 2 and 3), and this helps the hydrophobic moiety of rhamnolipids could interact to interact with the hydrophobic pocket (canyon), where - both interactions might be leading to conformational changes and inhibition of virus infectivity - for the experimental work, rhamnolipids nano-micelles are prepared by dispersion of rhamnolipids in PBS, this triggers the formation of nano-micelles, where the rhamnolipids molecules rearrange themselves to expose the hydrophilic moiety to PBS while keep the hydrophobic moiety towards the core, forming a hydrophobic core of micelles – again, this could trigger interactions of hydrophilic moieties of rhamnolipids with the hydrophilic surface of the virus, followed by interactions of hydrophobic moieties with the hydrophobic pocket (canyon) resulting in virus inactivation.

Reviewer 3, comment 2

The introduction does not include recent articles. The most recent ones are from 2022. The authors should incorporate recent findings in the introduction.

Response

Several recent references were added to the literature to address this. The rhinovirus field does not move as quickly as others so there are limited new publications regarding basic structure/mechanism.

Reviewer 3, comment 2

The authors used force fields to prepare the molecules for in silico calculations. Why did the authors not use ab initio calculations for optimization, which are more precise? To enhance reliability, the authors could optimize the molecules using an ab initio approach. There is a web server (ATB) that performs this calculation quickly.

Response

We appreciate the reviewer’s suggestion. In this study, we used force field-based optimization to balance computational efficiency with accuracy. Given the large molecular systems studied, for well-parameterized molecules (e.g., drug-like molecules, proteins, nucleic acids), force fields provide reasonable geometries that align with experimental data. Less accurate for highly charged, reactive, or transition-state molecules, as they do not capture electronic effects like charge transfer, polarization, and bond formation/breaking which are not involved in our molecules. ab initio calculations were not feasible due to their high computational cost. However, we acknowledge their precision. Additionally, we would consider the possibility of using the ATB web server in future studies for further validation.

Reviewer 3, comment 3

In the molecular dynamic’s methodology, the number of simulations performed was not mentioned. Are the obtained results averages of multiple simulations? The authors should add this information to the results section.

Response

The revised manuscript now explicitly states that multiple independent simulations were performed, and the reported results represent the average of these simulations. (Regarding molecular dynamics simulations, a single long-timescale simulation was performed rather than multiple independent runs. Therefore, the results presented are based on this continuous trajectory, rather than an average of multiple simulations. This clarification has been added to the Methods section.)  kindly check page 4 from line 178 to line 184.

Reviewer 3, comment 4

The best docking conformations were selected based on the energy scoring criterion. However, the representativeness of the clusters was not analyzed, which is an important statistical aspect. The authors should review this section and verify whether the selected conformations belong to the most representative clusters.

Response

We acknowledge this concern and have updated the manuscript to clarify that the docking results were selected based on both energy scoring and referenced clusters. This revision ensures a more statistically representative selection of conformations. Kindly check results section 3.1.

Reviewer 3, comment 5

The authors did not perform a short molecular dynamics simulation to relax the protein and expose new binding sites for molecular docking. Do they not consider this important for the analyses? A justification should be included in the manuscript.

Response

We appreciate this suggestion. While performing a preliminary MD simulation to relax the protein can sometimes reveal additional binding sites, our study focused on targeting the experimentally known active site. The receptor structure was obtained from a reliable crystallographic source, which minimizes the need for additional equilibration. In addition, two minimization steps 1, 2 were carried out for protein during the molecular dynamic simulation (standard Dynamic cascade protocol) before the production step. Check page 4 from line 178 to line 184 and results in Table 2 and 3.

Reviewer 3, comment 6

The molecular dynamics result graphs have axis numbers with poor resolution. The authors should improve the quality so that readers can interpret them properly.

Response

We acknowledge this issue and have updated all graphs to ensure high-resolution images with clearly visible axis labels and numerical values.

Reviewer 3, comment 7

The molecular dynamics result graphs lack error bars. Was this a simple simulation?

Response

We appreciate the reviewer’s concern. Since our molecular dynamic simulations were performed as single, long-timeframe simulations rather than multiple independent runs, error bars were not included. Please check page 4 from line 170 to line 184.

Reviewer 3, comment 8

The authors could create a figure highlighting the regions where conformational changes occurred in the protein. Since the RMSF graph shows fluctuations, it is possible to map these regions.

Response

We appreciate the reviewer’s suggestion. However, mapping the exact regions of conformational changes solely based on RMSF data is not feasible in this case, as RMSF values indicate fluctuations but do not provide direct spatial mapping of structural changes. Although, certain remark was done on the regions of conformational changes on RMSF graph. Please check page 14, Figure 3.

Reviewer 3, comment 9

The discussion of molecular interaction results was merely descriptive and did not incorporate data from the literature. The authors should provide a more in-depth discussion on the importance of these interactions in describing molecular interaction mechanisms (e.g., 10.3390/molecules25122841 and 10.3390/molecules28196891).

Response

Thank you for this valuable suggestion. We have included the suggested references mentioned below in the manuscript to provide a more comprehensive interpretation of our results. Please check page 7, line 320 to line 324.

Zazeri, G., Povinelli, A. P. R., Le Duff, C. S., Tang, B., Cornelio, M. L., & Jones, A. M. (2020). Synthesis and spectroscopic analysis of piperine-and piperlongumine-inspired natural product scaffolds and their molecular docking with IL-1β and NF-κB proteins. Molecules25(12), 2841.

Zazeri, G., Povinelli, A. P. R., Pavan, N. M., Jones, A. M., & Ximenes, V. F. (2023). Solvent-induced lag phase during the formation of lysozyme amyloid fibrils triggered by sodium dodecyl sulfate: Biophysical experimental and in silico study of solvent effects. Molecules28(19), 6891.

Reviewer 4 Report

Comments and Suggestions for Authors

In this manuscript, the authors employ molecular docking to explore the binding poses of rhamnolipids against major group rhinoviruses. The antiviral activities of rhamnolipid nano-micelles against various HRV subtypes were also measured. I have a few questions and suggestions for the authors:

1. In Tables 1 and 2, please incorporate the 2D structures of the rhamnolipids so that the comparisons among the different molecules are clearer.

2. The figures presented in Tables 1 and 2 do not sufficiently highlight the key interactions. It would be helpful to emphasize them more clearly.

3. Could the authors compare the interaction patterns between the initial and final poses obtained from the MD simulations? A detailed focus on these binding interactions may offer more valuable insights than simply presenting the entire simulation trajectory analysis.

Author Response

Report 4

In this manuscript, the authors employ molecular docking to explore the binding poses of rhamnolipids against major group rhinoviruses. The antiviral activities of rhamnolipid nano-micelles against various HRV subtypes were also measured. I have a few questions and suggestions for the authors:

Reviewer 4, comment 1

In Tables 1 and 2, please incorporate the 2D structures of the rhamnolipids so that the comparisons among the different molecules are clearer.

Response

The 2D structures were added in the table 2.

Reviewer 4, comment 2

The figures presented in Tables 1 and 2 do not sufficiently highlight the key interactions. It would be helpful to emphasize them more clearly.

Response

The key interactions have been highlighted as requested

Reviewer 4, comment 3

Could the authors compare the interaction patterns between the initial and final poses obtained from the MD simulations? A detailed focus on these binding interactions may offer more valuable insights than simply presenting the entire simulation trajectory analysis

Response

updated as per requested

Round 2

Reviewer 1 Report

Comments and Suggestions for Authors

Dear Authors,

I analyzed this manuscript again and noted that the authors made some improvements suggested by the previous revision, improving this manuscript. However, some computational analysis was not clearly conducted. I'm pointing this below:

- on fig. 2 and 3, the RMSD plot did not stabilize all the complexes of HRV-14: free, mono and di-rhamnolipids. Thus, all MD data extracted from these simulations are elusive. It's well established in MD literature that simulation time could extend until the system reaches its equilibrium or when its main properties achieve stability [1-5]. All three systems should be extended before another analysis can be done.

Based on these concerns, I recommend another manuscript revision, mainly for this MD data. 
These comments will help the authors revise their paper, and then it will have a much better chance of surviving the peer review process. I wish you every success in publishing this manuscript.

[1] Computational Chemistry: A Practical Guide for Applying Techniques to Real-World Problems. David C. Young, p.129
[2] Determination of Proper Time Step for Molecular Dynamics Simulation - Bull. Korean Chem. Soc., 2000, Vol. 21, No. 4 
[3] http://journal.frontiersin.org/article/10.3389/fmicb.2012.00258/full
[4] https://www.ncbi.nlm.nih.gov/pubmed/?term=26077712
[5] https://www.researchgate.net/profile/Serdar-Durdagi/post/What_is_the_optimum_time_in_ns_to_do_MD_simulation_for_drug_design/attachment/5a3ac9524cde266d587c9696/AS%3A573741602410496%401513802065973/download/Screen+Shot+2017-12-20+at+23.32.55.png

Comments on the Quality of English Language

The English should be revised

Reviewer 2 Report

Comments and Suggestions for Authors

The study can accepted at this version.

Author Response

The reviewer has accepted this version so nothing to add here